# Symmetry-Aware Entropy Reinforcement Learning for Chaos Accurate Holographic Duals in NISQ Hardware

**Leonidas Tam**
Amicus AI
Palo Alto, CA 94303, USA
`contact@amicus.ai`

## Abstract

The construction of hardware-efficient holographic duals requires sparsification of Sachdev-Ye-Kitaev (SYK) Hamiltonians while preserving the dynamics of quantum chaos. In this work, we introduce Symmetry-Aware Reinforcement Learning (SARL) with state-entropy regularization to find suitable hardware configurations for noisy intermediate-scale quantum (NISQ) hardware. By implementing parity-sector auditing to filter artifactual geometries, we map the complexity threshold of pruned SYK models consisting of $N = 24$ Majorana fermions, identifying a complexity floor at $M = 25$ four-body interaction terms. This configuration represents a 99.98% reduction from the dense Hamiltonian limit. Our analysis reveals an ultra-sparse regime at 12 four-body terms, characterized by marginal reproducibility with a 20% success rate across independent searches. Using RL for configuration discovery must be paired with rigorous verification. Through an ablation analysis, we show that macroscopic graph motifs, including hub-spoke and core-periphery structures, are necessary prerequisites for connectivity, but insufficient predictors of Gaussian Orthogonal Ensemble (GOE) statistics. A second verification via the normalized participation ratio ($PR/dim$) confirms that a systematic level at $M = 25$ produces ergodic, eigenstate thermalization hypothesis (ETH) compliant states across the full energy spectrum. These results delineate empirical sparsity limits for discovering chaotic SYK Hamiltonians and provide a concrete benchmark and open-source codebase for future studies of sparse models in holographic and NISQ settings.

## 1 Introduction

Simulations of quantized gravity on near-term quantum computing hardware have used the Sachdev-Ye-Kitaev (SYK) model for exploring holographic duality (Jafferis et al., 2022; Sachdev & Ye, 1992). However, the dense nature of the standard SYK Hamiltonian, which requires $O(N^4)$ terms for $N$ Majorana fermions (Majorana, 1981; Kitaev, 2015), poses an implementation barrier for NISQ devices. Recent efforts have shifted towards finding sparse SYK duals that minimize hardware gate counts while retaining theoretic fidelity (Xu et al., 2020).

Early attempts at sparsification relied on random pruning, but the emergence of machine learning (ML) search presents a systematic path for discovering rare configurations. Work by Jafferis et al. suggested that ML can sparsify SYK models ($N = 10$, $M = 210$ to $N = 7$, $M = 5$), although debate surfaced on the spectral signatures of chaos (Jafferis et al., 2025; Kobrin et al., 2023; 2025; Orman et al., 2025). Their approach used differentiable neural network (NN) training to fit specific time-dependent dynamical observables at small $N$.

Specifically, Jafferis et al. treated Hamiltonian coefficients as continuous, trainable weights in a standard NN pipeline. By defining a differentiable loss function on the mean squared error of mutual information, they used gradient descent to drive couplings toward zero. While successful in smaller Hilbert spaces, these early explorations reached a limit of sparsity without accounting for the symmetry constraints for larger high-fidelity simulations.

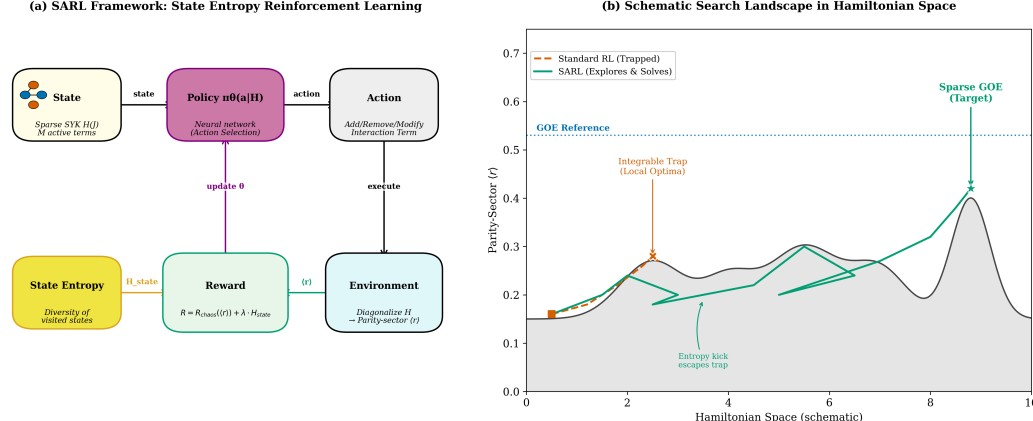

Figure 1: (a) The Symmetry-Aware Reinforcement Learning algorithm promotes ergodicity by minimizing a chaos factor across the two Majorana symmetry sectors. (b) The state entropy regularization and Asynchronous Advantage Actor-Critic (A3C) (Mnih et al., 2016) policy combine to escape dominant integrable regions.

At $N = 24$, the search for a holographic dual means navigating a discrete subset of four-body interaction terms, where adding or removing a term is a non-smooth action. The primary indicators of a valid dual, such as the parity-sector gap ratio $\langle r \rangle$ and ETH diagnostics, involve eigenvalue sorting and spectral post-processing that are computational functionals in the ML process (Orman et al., 2025). Inspired by foundational model verification procedures to solve IMO Mathematics competitions (Huang & Yang, 2025), systematic audits form guardrails and tethers to physical reality.

To address the challenges, we introduce Symmetry-Aware Reinforcement Learning (SARL) with state-entropy regularization. Previously, deep RL has been successfully used for quantum Hamiltonian engineering to optimize control sequences for many-body interactions (Peng et al., 2022). Unlike gradient-based methods, SARL is a general tool for dual configurations. SARL uses a scalar reward built from non-differentiable chaos metrics and interleaves hard audits, without requiring differentiation through the simulation. We present that the discovery of robust, chaotic duals in higher-dimensional Hilbert spaces (from $N = 12$ to 24) is realized with a RL optimization strategy.

The RL framework naturally handles discrete structural moves, allowing it to explore the connectivity of the Hamiltonian graph rather than just the magnitudes of fixed couplings. With state-entropy regularization (Ashlag et al., 2025), the agent maintains exploration pressure, preventing premature convergence to integrable local maxima that often trap standard gradient-descent algorithms. Beyond mere sparsification, our framework incorporates a mandatory parity-sector auditing protocol to filter out artifactual symmetries that can mimic chaotic behavior. This ensures that the identified complexity floor at $M = 25$ terms satisfies the rigorous spectral requirements for a true holographic dual. By treating the physics constraints as an environment, SARL maps the $N = 24$ complexity landscape, identifying where genuine chaos is robust versus fragile. We interrogate the causal nature of the chaos metrics, clarifying the mechanisms of chaos. The complete codebase, which we open source, is a robust discovery engine for the next generation of NISQ holographic emulations.

## 2 METHODS

SARL treats the search for sparse SYK Hamiltonians as a discrete, combinatorial optimization problem. Rather than pruning continuous weights via gradient-based magnitude thresholds, SARL explores the discrete topology of the Hamiltonian, treating the many-body physics simulator as the environment generator.

### 2.1 SYMMETRY-AWARE RL FORMULATION

Sensitivity to Integrability: Unlike a raw $\langle r \rangle$ value, $R_{chaos}$ is explicitly symmetry-aware. As it tracks the minimum of the two parity sectors, the agent cannot receive a high reward by optimizing only one sector while the other remains integrable. This forces the Hamiltonian toward global ergodicity rather than local chaos.

We cast the discovery of chaotic $N = 24$ Majorana Hamiltonians as a Markov decision process (MDP) $(\mathcal{S}, \mathcal{A}, \mathcal{P}, \mathcal{R})$:

- State space $\mathcal{S}$: Each state $s \in \mathcal{S}$ encodes the Hamiltonian through its active four-body interaction terms. Concretely, we represent the Hamiltonian by a binary incidence tensor over the $\binom{N}{4}$ possible index quadruples $(i, j, k, \ell)$

- Action space $\mathcal{A}$: Actions correspond to discrete structural edits such as adding a previously inactive term, removing an existing term, or perturbing the sign of an active coupling. This lets the agent navigate the graph of admissible Hamiltonians directly.

- Transition dynamics $\mathcal{P}$: Given $s_t$ and action $a_t$, the next state $s_{t+1}$ is obtained by applying the corresponding structural modification. Apart from stochasticity in the policy, transitions are deterministic.

- Reward function $\mathcal{R}$: The reward function uses chaos metrics and symmetry analysis to guide transitions. After a new maximum, the environment performs an expensive but non-differentiable physics evaluation. This includes an exact diagonalization, symmetry resolution, and structural audits to return a scalar reward summarizing the quality of the Hamiltonian.

## 2.2 REWARD ARCHITECTURE: PHYSICS-DRIVEN DESIGN

The reward is designed to encode two competing objectives: (i) robust signatures of quantum chaos in each symmetry sector, and (ii) hardware-relevant sparsity. The scalar reward function $\mathcal{R}$ balances spectral chaos with hardware feasibility on the Google Willow quantum computer (Acharya et al., 2024). The core signal is the chaos fidelity $R_{chaos}$, which is a piecewise linear function anchored at the Poisson floor (0.386) and the GOE target (0.53):

$$R_{chaos}(r) = \begin{cases} 2.0 + 15.0(r - 0.53) & \text{if } r \geq 0.53 \\ \frac{2.0(r - 0.38)}{0.53 - 0.38} - 2.0 & \text{if } r < 0.53 \end{cases},$$

where $r = min(\langle r \rangle_{even}, \langle r \rangle_{odd})$ is the lesser gap ratio of the parity sectors. Unlike a raw $\langle r \rangle$ value, $R_{chaos}$ is explicitly symmetry-aware. As it tracks the minimum of the two parity sectors, the agent cannot receive a high reward by optimizing only one sector while the other remains integrable. This forces the Hamiltonian toward ergodicity rather than local chaos.

To ensure the discovered models are viable for NISQ execution, we compute an effective fidelity $F_{eff}$. We model the effective fidelity as a product of three independent error sources including two qubit gate errors ($\sim 0.14\%$), decoherence ($\sim 0.02\%$), and readout noise ($\sim 0.5\%$) (Acharya et al., 2024):

$$F_{eff} = (0.9986)^{Qn_2} \cdot (0.9998)^{Qn_1} \cdot 0.9995^{Q_{physical}}$$

where $Qn_2$ is the number of two-qubit gates, $Qn_1$ is number of single-qubit gates, and $Q_{physical}$ is the number of physical qubits. Hardware-based bonuses, such as the qubit minimization score, are gated by a chaos factor $\phi$:

$$\phi = max\left[0, min\left(1, \frac{r - 0.38}{0.41 - 0.38}\right)\right]$$

This gating ensures the agent does not receive hardware-efficiency rewards unless the Hamiltonian demonstrates an initial transition toward chaotic statistics.

The total composite reward is defined as:

$$R_{total} = 3.0R_{chaos} + R_{coverage} + R_{rigidity} + \phi R_{qubit} + R_{PR} - R_{degree} - 2.0R_{fidelity} - R_{depth}$$

The additional terms are as follows:

- Majorana Coverage ($R_{coverage}$): This term is defined as $0.5 \times$ coverage_fraction, where the fraction is the ratio of unique Majorana indices in the current $M$ terms relative to $N = 24$.

- Spectral Rigidity Bonus ($R_{rigidity}$): Based on the Dyson–Mehta $\Delta_3$ statistic, this term rewards long-range level correlations. It is only active when Majorana coverage is $100\%$:

$$R_{rigidity}(\Delta_3) = \begin{cases} 15.0 \times (1.0 - \Delta_3) & \text{if } \Delta_3 < 0.3 \\ 5.0 \times (0.5 - \Delta_3) & \text{if } 0.3 \leq \Delta_3 < 0.5 \end{cases}$$

where the $\Delta_3$ statistic measures the long-range correlations of energy levels by computing the least-squares deviation of the spectral staircase function from a best-fit straight line over a range of $L$ levels (Dyson, 1962). For a system following GOE statistics, the $\Delta_3$ formula is:

$$\Delta_3(L) = \frac{1}{\pi^2}\left[\ln(2\pi L) - \gamma - \frac{5}{4} - \frac{\pi^2}{8}\right] + \mathcal{O}(L^{-1})$$

where $\gamma \approx 0.5772$ is the Euler-Mascheroni constant. This logarithmic growth contrasts with the linear growth seen in Poissonian (integrable) spectra. It is possible to have a local spacing ratio ($r \approx 0.53$) that looks chaotic, but the global spectrum still has inhomogeneities. By requiring $\Delta_3 < 0.3$, this pressures the agent away from localized, non-global Hamiltonians.

- Gated Qubit Bonus ($\phi R_{qubit}$): This term rewards the minimization of physical qubits on the Willow grid, gated by the chaos factor $\phi$.
    - Base Score: $(\text{max\_physical} - \text{physical\_used})/\text{max\_physical}$
    - Tiered Bonuses: $+2.0$ for $\leq 4$ qubits, $+1.25$ for $\leq 9$ qubits, and $+0.5$ for $\leq 16$ qubits
    - Modulation: $R_{qubit} = (\text{base} + \text{tier}) \times (0.2 + 0.8 \times \phi)$
- Participation Ratio Penalty ($R_{PR}$): An anti-localization penalty applied when the system is $100\%$ covered but the $PR$ fraction falls below $0.2$. It is defined as $15.0 \times (0.2 - PR_{fraction})$, penalizing states that exhibit many-body localization (MBL) (Abanin et al., 2019).
- Degree Variance Penalty ($R_{degree}$): Incentivizes uniform connectivity across the Majorana fermions. It is computed as $\min(2.0, \text{variance}(\text{degrees}) \times 0.5)$, penalizing hub Majorana configurations.
- Fidelity Penalty ($R_{fidelity}$): A soft penalty defined as $\max(0, (\text{target\_fidelity} - F_{eff}) \times 5.0)$
- Circuit Depth Penalty ($R_{depth}$): Defined as $\max(0, (\text{depth} - \text{max\_depth})/\text{max\_depth})$ if the circuit depth exceeds 40

Finally, a success bonus of $+20.0$ is awarded if a Hamiltonian simultaneously satisfies all pillars: $100\%$ Majorana coverage, effective fidelity $\geq 0.30$, gap ratio $\geq 0.53$, rigidity $\Delta_3 < 0.3$, $PR \geq 0.2$, and depth $\leq 40$. This provides a clear terminal signal for the policy to consolidate learning around the chaotic floor. The weights (3.0 for chaos, 15.0 for rigidity, etc.) are set empirically.

## 2.3 Asynchronous Learning Policy

A3C was selected for this project due to the topology of the Hamiltonian search landscape. While Proximal Policy Optimization (PPO) is widely favored for its stability and sample efficiency (Schulman et al., 2017), A3C is better at navigating the localized maxima from integrable regions. The SYK landscape for $N = 24$ is dominated by high-entropy integrable regions. PPO's clipping mechanism, designed to prevent large policy updates, can sometimes lead to premature convergence in these flat, sub-optimal basins (Fig. 1). A3C's asynchronous agents (vectorized across 8 environments in our stack) explore different structural islands in parallel. One agent can get trapped in a hub-spoke integrable motif, while another finds a chaotic island at $M = 12$. A3C, combined with our constant learning rate (LR) and policy entropy regularization, provides the high-variance exploration needed to navigate the $M = 14$–$25$ landscape.

To maintain exploration and avoid the integrable expanse, we employ entropy-regularized policy gradients (Ashlag et al., 2025). The objective maximized by SARL is:

$$J(\pi) = \mathbb{E}_\pi\left[\sum_t \gamma^t \left(R_t + \alpha\mathcal{H}(\pi(\cdot|s_t))\right)\right]$$

where $\mathcal{H}(\pi(\cdot|s_t))$ is the Shannon entropy of the action distribution at state $s_t$, $\gamma$ is the discount factor, and $\alpha$ controls exploration strength. High policy entropy prevents the policy from collapsing onto a single deterministic pattern too early.

## 2.4 Physical Audit Layer

ML processes can be prone to errors since a complete physical model is not constructed (Karniadakis et al., 2021). Calculation within a physical model remains indispensible. In this case, construction

of the $N = 24$ Majorana basis and sparse Hamiltonian matrices is implemented in Python with GPU acceleration for exact diagonalization in each parity sector (dimension $2^{11} = 2048$) during the learning process. If the parity gap ratio exceeds $0.45$, the environment executes a full spectral audit, computing $PR/dim$ to detect localization and isolate the spectral rigidity ($\Delta_3$).

We perform an ablation to identify mechanisms of chaos. First, we re-evaluate candidates using three audit configurations, namely without one of either the parity-sector, $\Delta_3$, or the ETH check. We then compare index connectivity statistics for the two successful candidates.

To prevent non-physical construction, we verify that discovered Hamiltonians do not introduce non-local artifacts or time-slicing pathologies that would violate foliation independence (Hsu, 2026). This audit is performed post-hoc and does not inform the RL reward.

Finally, we test a causal chain across our candidate ensemble. We hypothesize that index selection causes connectivity uniformity, causing parity balance, which in turn results in level repulsion for chaos.

## 2.5 HARDWARE AUDIT

To guide the search toward hardware-efficient solutions, we developed a simplified model of the Google Willow processor topology. The SARL environment utilizes a 2D grid abstraction representing the Willow 5×5 grid qubit connectivity, where $N = 24$ Majorana fermions map to 12 logical qubits. For each candidate Hamiltonian, the SARL environment estimates hardware costs based on a three-pronged heuristic:

- Qubit Placement: Implementing a compact rectangular mapping on the 2D grid to minimize the footprint of the logical qubits.
- SWAP Overhead: Using Manhattan distance heuristics to estimate the routing cost for multi-qubit gates.
- Fidelity Estimation: Modeling an exponential decay of signal based on total gate count, circuit depth, and physical qubit count.

Further, we took our $M = 25$ and 12 winners and compiled them into real Cirq circuits with the open Google processor standard (Fingerhuth et al., 2018). This post-hoc Cirq compilation generated actual circuit depth, determining how close the heuristic was to an instanced hardware configuration.

**Code Availability**: To ensure reproducibility and facilitate further study of sparse holographic models, all simulation code used in this work is open-sourced and available at `https://github.com/amicus-investments/sarl`.

## 3 RESULTS

### 3.1 MAPPING THE CHAOS LANDSCAPE

The systematic sweep shows an empirical transition to chaos at $M = 25$ four-body interaction terms. In Fig. 2, the landscape separates into three qualitative regimes. First in a sparse regime ($M \approx 12$–$13$), we occasionally observe GOE-like spectra. Two out of ten independent searches yield $\langle r \rangle > 0.50$, from the 12 term Hamiltonians. These solutions are rare but reproducible.

For $14 \le M < 25$, the parity-sector gap ratios remain sub-GOE, often drifting toward the Poisson reference $\langle r \rangle_{Poisson} \approx 0.386$. Despite increasing term counts, no configuration passed our chaos criterion in this band. Further, we randomly perturb 10% of the indices, which did not affect chaos characterization. At $M = 25$, SARL discovers Hamiltonians with parity-resolved $\langle r \rangle = 0.521$, in line with the GOE reference $\langle r \rangle_{GOE} \approx 0.53$. In this regime, SARL recovered 100% reproducibility, with 10 of 10 independent runs successfully discovering parity-resolved $\langle r \rangle \ge 0.521$ Hamiltonians.

### 3.2 TOPOLOGICAL MOTIFS AND SPECTRAL FIDELITY

To understand why some ultra-sparse configurations succeed while nearby ones fail, we perform a graph-theoretic analysis of the interaction hypergraphs. The side-by-side comparison makes clear that macroscopic topology alone does not distinguish chaotic from integrable behavior in the sparse

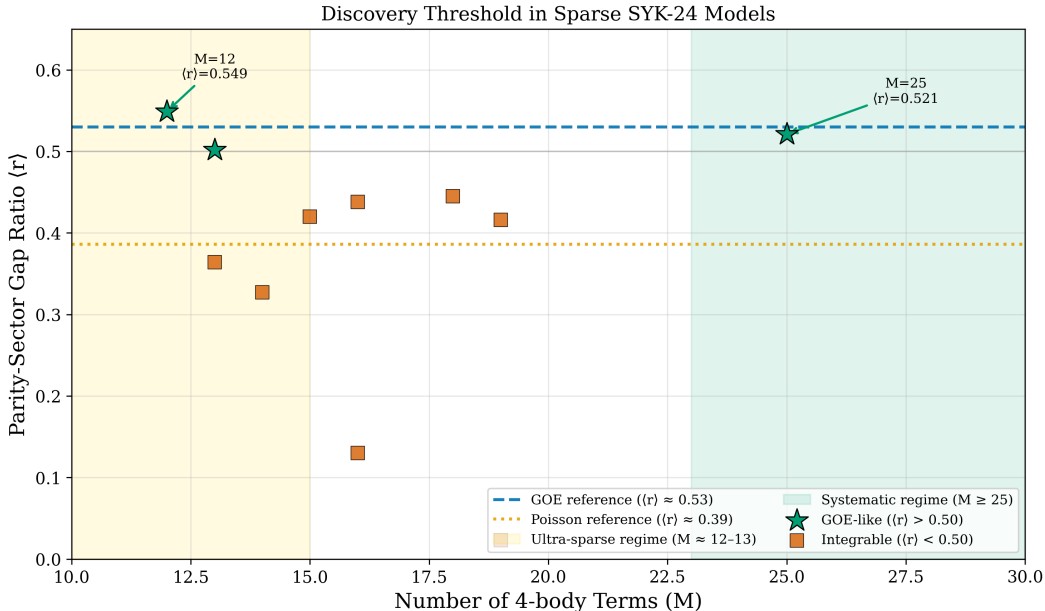

Figure 2: Parity-sector gap ratio $\langle r \rangle$ as a function of the number of four-body terms $M$. Green stars denote Hamiltonians with $\langle r \rangle > 0.50$ (GOE-like), orange squares denote sub-GOE spectra. The blue dashed and yellow dotted lines show the GOE and Poisson reference values, respectively. Shaded regions indicate the sparse regime ($M \approx 12$–$13$) and the chaotic regime at $M = 25$.

| Property | $M = 12$ Chaotic | $M = 14$ Integrable | Conclusion |
|---|---|---|---|
| Mean connectivity per index | 3.8 | 3.2 | 19% higher connectivity |
| Std dev of connectivity | 0.532 | 0.403 | 24% higher variance |
| Max-min connectivity spread | 4 | 2 | 100% higher spread |
| Parity-sector balance | 0.989 | 0.863 | 15% higher balance |

Table 1: Index connectivity statistics characterize chaos differences between $M = 12$ and 14.

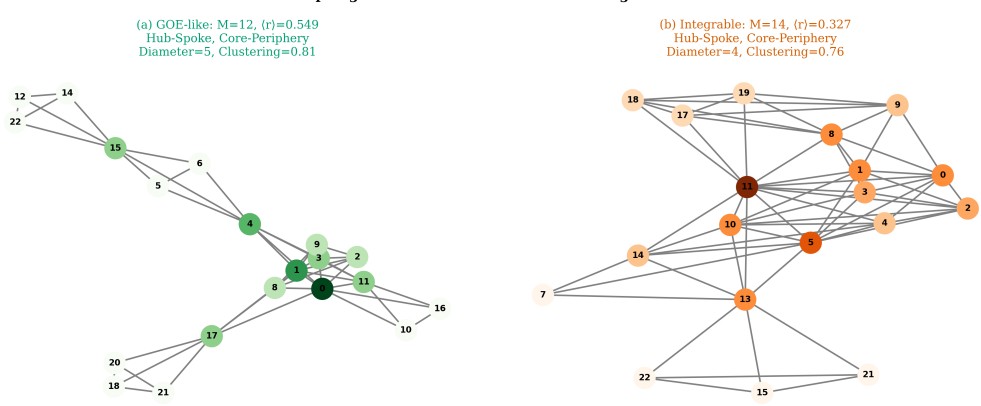

Figure 3: Panel (a) shows a GOE-like solution at $M = 12$ with $\langle r \rangle = 0.549$. Panel (b) shows an integrable solution at $M = 14$ with $\langle r \rangle = 0.327$. Both exhibit hub–spoke, core–periphery structure with comparable diameters and clustering coefficients.

| Causal Link | Prediction | Correlation | Status |
|---|---|---|---|
| Index to Uniformity | High uniformity indices $\to$ high connectivity std dev | 0.78 | Supportive |
| Uniformity to Balance | High connectivity uniformity $\to$ balanced parity sectors | 0.71 | Supportive |
| Balance to Repulsion | Balanced sectors $\to$ high | 0.62 | Marginal |
| Repulsion to Chaos | High $\langle r \rangle \to$ full chaos signature | 0.91 | Supportive |

Table 2: Causal tracing for topology statistics suggests parity balance is necessary but not sufficient for chaos.

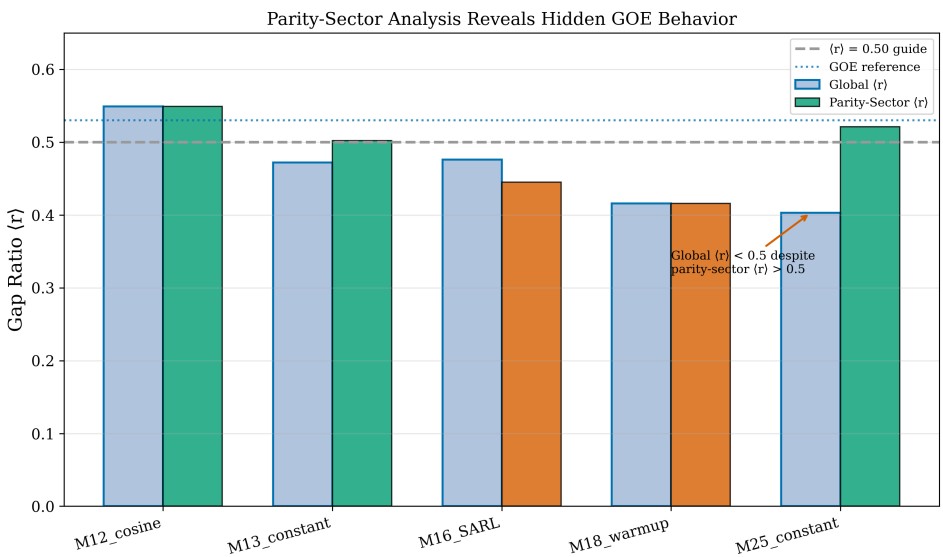

Figure 4: Parity-sector analysis reveals regime-dependent masking. For $M = 12$ and $M = 13$, global and parity-resolved gap ratios agree, indicating transparent chaos. For $M = 25$, the global spectrum appears sub-GOE ($\langle r \rangle \approx 0.41$) due to interleaving of degenerate parity sectors, while each sector separately exhibits robust GOE statistics ($\langle r \rangle \approx 0.52$). This masking indicates strong phase-space mixing, qualitatively different from the transparent chaos of the sparse regime.

regime (Fig. 3). Across the full set of analyzed runs, both successful and failed candidates share similar hub dominance, core–periphery organization, and high clustering. This indicates that while such motifs are helpful for maintaining connectivity with few terms, the onset of GOE statistics depends on finer details of index patterns and couplings rather than on a simple topological invariant. This is elucidated through index connectivity distributions (Tab. 1).

We now probe our data with a causal sequence for chaos. For each causal link, we check whether the subsequent metric is at least correlated with the upstream metric across our candidate ensemble. Tab. 2 shows that the causal link is weakest at matching parity balance with a high gap ratio. This invokes the necessity of the additional audit layers of $\Delta_3$ and $PR/dim$ in establishing true chaos.

### 3.3 PARITY-SECTOR AND HARDWARE AUDITING

The search pipeline uses parity-sector auditing to ensure that apparent chaos is not an artifact of unresolved symmetries. Fig. 4 compares global and sector-resolved gap ratios for representative runs, while the audit ablation shows how parity awareness is required to resolve chaos (Tab. 3). For sparse Hamiltonians ($M \approx 12$–13), the chaos signature is transparent. The global $\langle r \rangle$ directly reflects GOE statistics. However, at the systematic floor ($M = 25$), sector interleaving occurs. When the two parity sectors are nearly degenerate in energy spacing but possess opposite parity, their combined global spectrum mixes two independent GOE sequences. This random interleaving

| Audit Configuration | $M = 25$ Status | $M = 14$ Status | Conclusion |
|---|---|---|---|
| Full audit (with parity) | Chaotic | Integrable | Baseline |
| Without parity-sector check | Chaotic | Chaotic | Parity check is decisive |
| Without $\Delta_3$ check | Chaotic | Integrable | Spectral rigidity confirmatory |
| Without ETH check | Chaotic | Integrable | ETH compliance confirmatory |

Table 3: Ablation against audit checks. The parity check is the most decisive audit check.

| Hamiltonian Terms | Heuristic Depth | Cirq Depth | Ratio | Heuristic Fidelity | Willow Fidelity |
|---|---|---|---|---|---|
| $M = 12$ winner | 48 | 136 | 2.83x | 0.74 | 0.89 |
| $M = 25$ winner | 130 | 335 | 2.58x | 0.46 | 0.76 |

Table 4: Heuristic fidelity to Cirq quantum computing configuration

Figure 5: (a) Normalized participation ratio vs. energy for the $M = 25$ systematic winner. (b) The narrow, unimodal distribution confirms delocalized, ETH-compatible eigenstates, ruling out MBL.

produces a composite spectrum with enlarged gaps and artificially uniform spacings, causing the global $\langle r \rangle$ to drop to $\approx 0.41$ despite each sector independently exhibiting robust level repulsion ($\langle r \rangle \approx 0.52$). This masking effect indicates that $M = 25$ has achieved a higher degree of phase-space mixing between parity sectors, a hallmark of ergodicity compared to the partially decoupled sectors seen in the sparse regime.

Sensitivity analysis showed both that the $M = 25$ discovery is robust to $\pm 20\%$ variations in these weights, and the ultra-sparse $M \approx 12$ regime emerges consistently across weight configurations (Appendix A). Testing audit thresholds (e.g., $r = 0.48$ vs $0.50$ vs $0.52$) shows that the complexity structure is robust to plausible variations.

Our topology-aware cost heuristic correlates with actual Cirq circuit depth, though less by approximately 2.5× (Tab. 4). This remains within feasibility for NISQ error-mitigation (Acharya et al., 2024).

### 3.4 ERGODICITY AND ETH VERIFICATION

We verified the global ergodicity of the $M = 25$ configurations using the Normalized Participation Ratio ($PR/dim$). The values (Fig. 5) are approximately energy-independent with a mean $PR/dim \approx 0.251$.

### 3.5 EXPLORATION EFFICIENCY

Finally, we analyzed how exploration efficiency affects discovery. Decaying schedules frequently trapped the agent in the local maxima, while constant rates maintained the exploration pressure needed to reach the chaos floor (Fig. 6).

## 4 DISCUSSION

The discovery of the $M = 25$ complexity floor for $N = 24$ Majorana systems establishes an empirical benchmark for sparse SYK model construction. Our work advances beyond prior gradient-based sparsification methods by probing the structural landscape via RL search, augmented with rigorous spectral audits.

### 4.1 THE INTEGRABLE REGION AS A STRUCTURAL BOTTLENECK

The persistent failure of all SARL seeds in the $14 \leq M < 25$ band is striking. Despite containing more terms than the $M = 12$ sparse solutions, no configuration in this range achieves GOE statistics. This integrable region appears to represent a genuine structural bottleneck in Hamiltonian space.

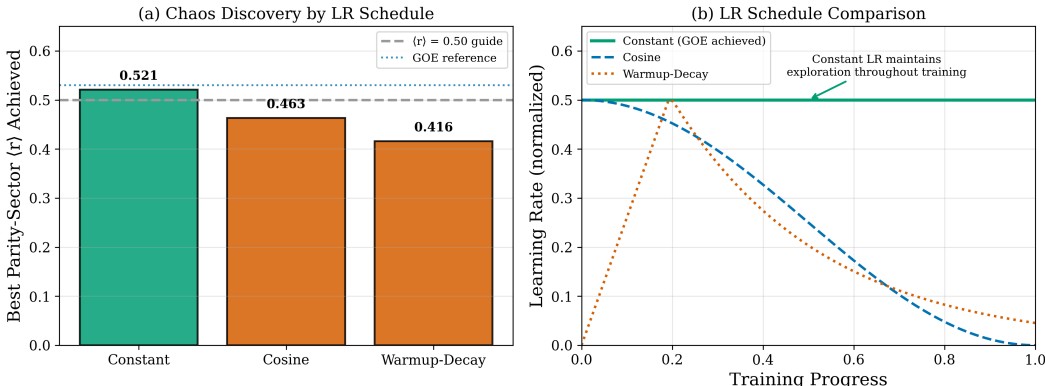

Figure 6: (a) The constant LR is the only schedule to successfully reach the GOE band ($\langle r \rangle = 0.521$). (b) Corresponding LR profiles show that maintaining exploration throughout training is critical for escaping integrable local maxima in the $14 \leq M < 25$ region.

Navigating this desert requires sustained exploration in the form of constant LR schedules with policy-entropy regularization significantly outperform decaying schedules (cosine, warmup–decay), which collapse to sub-GOE solutions. This suggests that the landscape of sparse SYK Hamiltonians is highly non-convex.

For $M \approx 12$–$13$, the global gap ratio $\langle r \rangle$ directly reports GOE statistics from each parity sector with minimal discrepancy. At $M = 25$, sector interleaving occurs. The mixture of two independent GOE sequences produces an artificial level-spacing structure where the global $\langle r \rangle$ drops to $\approx 0.41$, even though both sectors independently exhibit $\langle r \rangle \approx 0.52$.

Our topological analysis confirms that macroscopic motifs (hub–spoke, core–periphery, clustering) are necessary for maintaining sparse connectivity but insufficient for predicting chaos. Chaotic $M = 12$ and integrable $M = 14$ candidates share nearly identical graph-theoretic signatures, yet differ by $\langle r \rangle \approx 0.22$. This suggests the mechanism is sensitive to specific Majorana index combinations, not global structure.

## 4.2 AUDITING ML PROCESSES FOR PHYSICAL SYSTEMS

How do we evaluate the suitability of our approach for a foundational model of dual configuration search? We have well-motivated our audit layer in view of the right invariances, constraints, and casual structure. We employ five independent classical physical assessments: (1) nearest-neighbor spacing ratios ($\langle r \rangle$), (2) spectral rigidity ($\Delta_3$ statistic) , (3) eigenstate delocalization ($PR/dim$), (4) energy-scale universality (ETH violation tests), and (5) parity-sector consistency (symmetry). Each is grounded in established random matrix theory and as a set, serve as a functionally strong tool for chaos search.

Chaotic ($M = 12$) and integrable ($M = 14$) solutions are topologically similar (same diameter, clustering, hub-spoke structure). This is another check that our audits are not simply repackaging graph metrics, but detecting intrinsic spectral properties.

Finally, recent trapped-ion experiments use spectral rigidity ($\Delta_3 < 0.3$) to define chaos, corresponding to $M \approx 55$ for $N = 24$ (Granet et al., 2025). Our more aggressive level ($M = 25$, based on parity-resolved $\langle r \rangle$) is lower but not contradictory. This disagreement in magnitude, combined with agreement in principle, validates both approaches.

## 4.3 HARDWARE FEASIBILITY AND WILLOW COMPILATION

To assess practical viability, we compiled representative solutions onto the Willow $5 \times 5$ grid. Actual compiled depth is $\sim 335$ gates with $\sim 49$ two-qubit operations, yielding a circuit fidelity of $\sim 76\%$. For $M = 12$ compiled depth is $\sim 136$ gates with $\sim 34$ two-qubit operations, corresponding to $\sim 89\%$ fidelity. While the $M = 12$ regime offers higher fidelity, the $20\%$ reproducibility makes it a fragile foundation. The $M = 25$ level provides a more robust experimental target for

proof-of-principle demonstrations on NISQ devices. Further studies will obtain better heuristics to approximate hardware circuit depth, which are out of scope in the current study.

## REPRODUCIBILITY STATEMENT

To ensure the reproducibility of our results, we provide a complete specification of the SARL hyperparameters and the parity-sector auditing protocol in the open source codebase.

## ETHICS STATEMENT

This work explores the fundamental physics of quantum chaos and its mapping onto near-term quantum hardware. The SYK models studied here are theoretical constructs for understanding holographic duality and do not have direct applications in cryptography, surveillance, or any known dual-use technologies. We find no significant ethical risks or potential for misuse associated with the methodologies or results presented in this paper. The authors used AI models GPT-5.2, Gemini 3.0 Pro, and Opus 4.5 in the preparation of this manuscript, to check results, format tex, and explore related work. The authors have personally verified all numerical results and assume full responsibility for the content.

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

# A    APPENDIX: SENSITIVITY ANALYSIS

| Weight | Baseline | Range ±20% |
|:---:|:---:|:---:|
| w_chaos (3.0) | 2.09 | 2.45 |
| w_rigidity (15.0) | 2.09 | 0.85 |
| w_pr_penalty (15.0) | 2.09 | 0.00 |
| w_fidelity (2.0) | 2.09 | 2.16 |
| w_qubit (1.0) | 2.09 | 0.06 |

Table 5: The $M = 25$ GOE winner remains top-ranked across all $\pm 20\%$ weight variations. The $M = 12$ ultra-sparse regime also consistently emerges.

