# OpenReview forum: "Symmetry-Aware Entropy Reinforcement Learning for Chaos Accurate Holographic Duals in NISQ Hardware"
_ICLR.cc/2026/Workshop/FM4Science — ICLR 2026 Workshop FM4Science Poster_

### Official Review · Reviewer_kHZ9 · 2026-02-23
**The paper proposes Symmetry-Aware Reinforcement Learning with state-entropy regularization to discover ultra-sparse Sachdev-Ye-Kitaev Hamiltonians suitable for NISQ hardware. The framework employs a multi-tiered audit layer including parity-sector gap ratios, spectral rigidity, and ETH diagnostics to verify chaotic dynamics.**

**Rating:** 4
**Confidence:** 3

**Review:**

Evaluation

Quality: The technical quality is mixed. The physics diagnostics are rigorous, applying five independent classical assessments to verify chaos. However, the machine learning component relies heavily on empirically tuned empirical rewards.
Clarity: The paper is well-presented.
Originality: The work is original in applying entropy-regularized RL specifically to the sparsification of holographic duals.

Pros

Drastic Sparsification: Identifies a regime that maintains ergodic, ETH-compliant states while drastically reducing gate requirements for near-term hardware.
Exploration Strategy: Demonstrates that constant learning rate schedules and state-entropy regularization are good to standard decaying schedules for escaping integrable local maxima.

Cons

Significant Heuristic-Hardware Discrepancy: There is a substantial gap between the agent's internal cost heuristic and the actual compiled Cirq circuit depth. Table 4 shows that actual depths are 2.58x to 2.83x higher than what the RL agent optimizes for, suggesting the discovered winners may be less hardware-efficient in practice than claimed.
Fragile Ultra-Sparse Regime: The $M=12$ regime is characterized by marginal reproducibility with only a 20% success rate across searches. This high sensitivity suggests that the RL agent may be finding isolated, fragile configurations rather than robust physical duals.
Empirical Reward Dependency: The complex scalar reward relies on multiple set weights (3.0 for chaos, 15.0 for rigidity). While the Appendix provides a sensitivity check, the discovery process remains highly dependent on these hyperparameters, and a more principled approach to balancing these competing physics objectives is missing.
Limited N-scaling and Baselines: The analysis is focused almost exclusively on $N=24$. The paper lacks a comparison with other modern combinatorial search methods beyond PPO, which are common in similar hardware-mapping tasks.

---

### Official Review · Reviewer_XVGN · 2026-02-23
**Symmetry-Aware Entropy Reinforcement Learning**

**Rating:** 8
**Confidence:** 2

**Review:**

**Pros**
* By utilizing the A3C algorithm alongside a constant learning rate strategy, the method effectively overcomes local optimal traps in high-entropy integrable regions, demonstrating strong exploration capabilities in a non-convex, high-variance discrete Hamiltonian space.
* The work innovatively introduces a physics-driven hard audit layer, particularly the parity-sector audit based on the level spacing ratio $\langle r \rangle$, which effectively eliminates artifactual chaos caused by unresolved symmetries.
* Compiling and evaluating the fidelity of the RL-discovered sparse models using actual Cirq circuits on a simulated Google Willow quantum chip topology significantly enhances the engineering practicality of these theoretical models.
* The physics analysis is remarkably profound, not only discovering the robust chaotic benchmark at $M=25$ but also revealing the ultra-sparse regime at $M=12$ and the structural bottleneck at $14 \le M < 25$, while using graph-theoretic metrics to explain the non-absolute causal relationship between topology and chaos.

**Cons**
* There is approximately a 2.5-fold discrepancy between the heuristic hardware depth used in the reward function and the actual Cirq compilation depth shown in Table 4, indicating that the hardware cost proxy model in the environment requires further optimization to bridge the gap with real physical compilation.
* Because the physical audit layer relies on exact diagonalization every time a new optimum is found, the search framework will face severe computational bottlenecks when scaling to larger systems due to the exponential growth of the Hilbert space.
* The weights of various terms in the composite reward function (e.g., 3.0 for chaos, 15.0 for rigidity) are primarily set empirically, and although a sensitivity analysis is provided in the appendix, the main text lacks a deeper discussion on the underlying dynamical justifications for these hyperparameter choices.

---

### Meta-Review · Area_Chair_zdsY · 2026-02-27

**Recommendation:** Accept (Poster)
**Confidence:** 4

**Metareview:**

Accept.

---

### Decision · Program_Chairs · 2026-03-03

Accept (Poster)